# Bacterial virulence plays a crucial role in MRSA sepsis

**Gordon Y. C. Cheung, Justin S. Bae** **, Ryan Liu, Rachelle L. Hunt** ¤**, Yue Zheng, Michael Otto** *

Pathogen Molecular Genetics Section, Laboratory of Bacteriology, National Institute of Allergy and Infectious Diseases, National Institutes of Health, Bethesda, Maryland, United States of America

¤ Current address: Yale University, New Haven, Connecticut, United States of America
* motto@niaid.nih.gov

## Abstract

Bacterial sepsis is a major global cause of death. However, the pathophysiology of sepsis has remained poorly understood. In industrialized nations, *Staphylococcus aureus* represents the pathogen most commonly associated with mortality due to sepsis. Because of the alarming spread of antibiotic resistance, anti-virulence strategies are often proposed to treat staphylococcal sepsis. However, we do not yet completely understand if and how bacterial virulence contributes to sepsis, which is vital for a thorough assessment of such strategies. We here examined the role of virulence and quorum-sensing regulation in mouse and rabbit models of sepsis caused by methicillin-resistant *S. aureus* (MRSA). We determined that leukopenia was a predictor of disease outcome during an early critical stage of sepsis. Furthermore, in device-associated infection as the most frequent type of staphylococcal blood infection, quorum-sensing deficiency resulted in significantly higher mortality. Our findings give important guidance regarding anti-virulence drug development strategies for the treatment of staphylococcal sepsis. Moreover, they considerably add to our understanding of how bacterial sepsis develops by revealing a critical early stage of infection during which the battle between bacteria and leukocytes determines sepsis outcome. While sepsis has traditionally been attributed mainly to host factors, our study highlights a key role of the invading pathogen and its virulence mechanisms.

## Author summary

Bacterial infections often develop sepsis as a complication. Sepsis is a severe blood infection and one of the main reasons for death, especially in hospitals of the developed world. Sepsis is believed to be due to an overshooting immune reaction to structures that most bacteria share. However, this model fails to explain why some bacteria cause more frequent and severe sepsis than others. In our study, which we performed with the main sepsis-causing bacteria, *Staphylococcus aureus*, we show that the outcome of sepsis depends on the battle between bacteria and white blood cells that happens early during infection. Many of the weapons that bacteria use for that battle are controlled by a quorum-sensing regulator, which implies that so-called anti-virulence strategies directed at that regulator

**Data Availability Statement:** All relevant data are within the manuscript and its Supporting Information files.

**Funding:** This study was supported by the Intramural Research Program of the National

Institute of Allergy and Infectious Diseases (NIAID), U.S. National Institutes of Health (NIH), project number ZIA AI000904 (to M.O.). The funder had no role in study design, data collection and analysis, decision to publish, or preparation of the manuscript.

**Competing interests:** The authors have declared that no competing interests exist.

may work to treat sepsis. However, we also show that targeting quorum-sensing is counterproductive when sepsis originates from biofilms on implanted catheters, which it often does. Our study shows that bacterial weaponry plays a key role in sepsis and gives important advice on how to use this finding for alternative drug development.

## Introduction

Sepsis results from bacteremia, the invasion of bacteria into the bloodstream, and represents one of the most frequent causes of death. Globally, an estimated ~ 50 million cases of sepsis are reported with ~ 11 million fatalities [1]. In the United States, sepsis is the predominant cause of death in hospitalized patients and costs the healthcare system ~ $ 24 million annually [2,3].

Sepsis is a life-threatening organ dysfunction syndrome that traditionally is believed to result from an overwhelming and dysregulated immune response to bacterial intruders [4–6]. One of the hallmarks of the inflammatory septic response is the production of specific pro-inflammatory cytokines, such as IL-6 or TNF-α, while anti-inflammatory cytokines, such as IL-4, are also produced to keep the inflammatory reaction under control [7]. However, the pathophysiology of sepsis is still not completely understood and there is considerable controversy regarding the role of immune suppression during sepsis [8–13] and whether cytokines are appropriate sepsis biomarkers [14–17].

Phylogenetically widely conserved bacterial surface structures, such as lipopolysaccharide (LPS) in Gram-negative and lipoteichoic acid and lipopeptides in Gram-positive bacteria, are commonly regarded as the primary triggers of the host inflammatory response to bacterial infection [18,19]. However, this model cannot explain why some bacteria cause more severe and frequent cases of sepsis than others; and more recent reports have suggested a role of toxin-mediated bacterial virulence in the progression and outcome of sepsis [20–22].

*Staphylococcus aureus* is the leading cause of blood infections in industrialized nations and the pathogen causing the highest mortality [23,24]. Antibiotic resistance as present in methicillin-resistant *S. aureus* (MRSA) leads to worsened clinical outcome of staphylococcal bacteremia [25]. Importantly, most MRSA blood infections originate from infections on indwelling medical devices, such as in the case of intravascular catheter-related *S. aureus* bacteremia [26,27], and characteristically involve biofilm formation on the infected devices [28].

The quorum-sensing system Agr (accessory gene regulator) controls most virulence determinants in *S. aureus*, including toxins that lyse leukocytes and other immune evasion factors [29]. For that reason and in the light of frequent antibiotic resistance in *S. aureus*, quorum-quenching approaches that target Agr functionality are often proposed as potential alternatives to antibiotic-based treatment of *S. aureus* infection [30]. However, clinical and experimental data regarding the impact of Agr quorum-sensing on sepsis are somewhat contradictory. Most [31–34], yet not all [35], studies using isogenic *S. aureus agr* mutants in experimental bacteremia indicate a strong impact of Agr on mortality. Then again, there is increased incidence and association with disease severity of naturally occurring functionally Agr-deficient isolates from clinical cases of bacteremia [36–38]. What further complicates the matter is the fact that Agr also impacts biofilm formation via strict control of the phenol-soluble modulin (PSM) surfactant peptides [39,40], which leads to increased biofilm extension and resistance to immune clearance of Agr-deficient strains during infection of indwelling medical devices [41–43].

Thus, as is the case for most bacteria causing bloodstream infections, the role of virulence and quorum-sensing in *S. aureus* sepsis remains incompletely understood [44]. This is to a large part due to the lack of detailed investigation of the events accompanying bacterial sepsis,

such as interaction with leukocytes and cytokine production. Such investigation–particularly during early stages of sepsis—is hardly possible in the clinic and requires animal models. Furthermore, despite recently obtained insight into the role of quorum-sensing in *S. aureus* biofilm-associated infection [29,41,42,45], biofilm-associated infection models have not yet been used to investigate the role of quorum-sensing in sepsis as its main clinically important complication. As a consequence, the often-claimed potential of quorum-sensing blockers for systemic *S. aureus* and MRSA infection lacks thorough experimental confirmation.

Here, to understand the role of virulence and quorum-sensing in *S. aureus* sepsis, including sepsis of device-associated origin, we performed animal models of non-catheter-associated and catheter-associated sepsis using the MRSA strain USA300 (LAC), the most frequent cause of *S. aureus* infections in the Unites States [46]. In contrast to most previously published *S. aureus* infection studies, which were performed in mice, we focused on a rabbit model for the following reasons. First, only use of a larger animal allows repeated drawing of sufficient blood to monitor the development of sepsis-related parameters. Second, mice–in contrast to rabbits—are not sensitive to several *S. aureus* leukocidins [47], making rabbits a better choice for the analysis of the general impact of Agr, Agr-regulated virulence factors, and overall *S. aureus* virulence during systemic infection.

Our results demonstrate a critical role of bacterial virulence, and particularly of *S. aureus*-neutrophil interaction, for the outcome of sepsis and highlight fundamental differences between device- and non-device-related origins of sepsis regarding the role of the impact of quorum-sensing. Overall, they discourage the use of quorum-quenching but underline the potential of approaches directly targeting leukocyte-bacteria interaction for the treatment of *S. aureus* sepsis.

## Results

### The quorum-sensing virulence regulator Agr strongly impacts mortality in a mouse sepsis model

Mouse models of acute, systemic *S. aureus* infection that produce a sepsis scenario and pronounced mortality within days post infection have frequently been used before. This includes studies that compared *agr* mutants and corresponding isogenic parental strains [31–34]. To study the role of virulence and quorum-sensing in *S. aureus* sepsis, we here also first performed a mouse study comparing mortality due to infection with wild-type LAC versus isogenic Δ*agr* bacteria. Additionally, we tested the impact of cyclophosphamide (CY), which depletes animals of leukocytes [48]. This analysis specifically addresses the impact of *S. aureus* virulence, because (i) phagocytosis by leukocytes, and among them particularly neutrophils, is known to represent the primary factor controlling *S. aureus* infection [49] and (ii) a wide array of *S. aureus* cytolytic toxins that target leukocytes, such as the bicomponent leukocidins, α-toxin and PSMs, are Agr-controlled and represent major virulence determinants of *S. aureus*, as demonstrated by the significant effects they have on the outcome of *S. aureus* infection [47,50–52].

Mice inoculated with equal amounts of the *S. aureus* wild-type strain LAC succumbed to infection much more rapidly than those infected with the isogenic Δ*agr* mutant (p<0.0001) (**Fig 1A**). In CY-treated mice, in contrast to non-treated mice, there was no significant difference in mortality between wild-type and Δ*agr*-infected animals (**Fig 1B**), confirming the idea of a major impact of bacteria-induced leukocyte lysis on mortality in this model. These results indicate that bacterial virulence and resulting leukocyte numbers have a determining impact on the outcome of *S. aureus* sepsis.

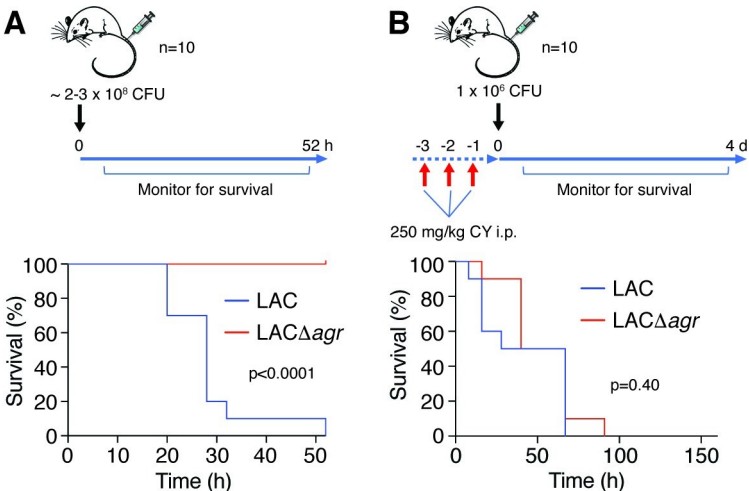

**Fig 1. Mouse sepsis model.** (**A**) Scheme of experimental setup and survival experiment. Survival of female C57BL/6NCrl mice following intravenous challenge with ~2–3 × 10$^8$ CFU of *S. aureus* LAC or its isogenic *agr* deletion strain (n = 10/group) was recorded. Animals were monitored for survival up to 52 hours. Statistical analysis is by log-rank (Mantel-Cox) test. (**B**) Survival experiment under cyclophosphamide (CY) treatment. Scheme of experimental setup and survival data. Survival of female C57BL/6NCrl mice following intravenous challenge with 10$^6$ CFU of *S. aureus* LAC or its isogenic Δ*agr* deletion strain (n = 10/group) was recorded. Statistical analysis is by log-rank (Mantel-Cox) test. Mouse picture is from openclipart.org.

## The impact of Agr on sepsis-related mortality is strongly dependent on whether the infection is device-associated

As already mentioned, the mouse is a suboptimal model organism to study the impact of virulence on sepsis caused by *S. aureus*. We therefore performed all following experimentation in rabbits to confirm and further elaborate on the findings we had achieved in mice. In these experiments, we included a catheter-associated model to assess the specific impact of device-associated infection as the most frequent type of staphylococcal blood infection.

In a non-catheter-related infection setup, essentially reproducing the above mouse model in rabbits (**Fig 2A**), we observed a slightly increased mortality rate in animals infected with the wild-type as compared to those infected with the Δ*agr* strain (**Fig 2B**). This difference was only statistically significant in the Gehan-Breslow-Wilcoxon analysis, which attributes more weight to early time points (p = 0.038), but not in the usual log-rank test (p = 0.15). In contrast, in a setup with an inserted silastic central venous catheter (CVC), the Δ*agr* strain produced significantly higher mortality using both statistical analysis methods (p = 0.0011, log-rank; p = 0.0013, Gehan-Breslow-Wilcoxon) (**Fig 2C**). CFU over the time course of infection in the blood and organs reflected the mortality results, inasmuch as they were (i) increased in moribund animals, at least in the blood and some organs, (ii) similar between wild-type and Δ*agr*-infected mice in the non-catheter-associated model, and (iii) overall higher in Δ*agr*-infected rabbits in the catheter-associated model (**Figs 3A, 3B, 4A and 4B**). There were no apparent differences in weight loss but we observed decreased rectal temperature in moribund animals (**S1 Fig**). Notably, most (10/12) rabbits infected with the Δ*agr* strain developed biofilms on the device and of these rabbits, almost all (9/10) showed high blood and organ CFU and succumbed to the infection, whereas only one (1/12) rabbit infected with the wild-type strain died (**Fig 4A and 4C**). (The latter finding contrasts the results in the non-catheter-associated model considering that the same inoculum was used, but is probably to be explained by the catheters sequestering the bacteria and thus initially removing them from the bloodstream.) Only one

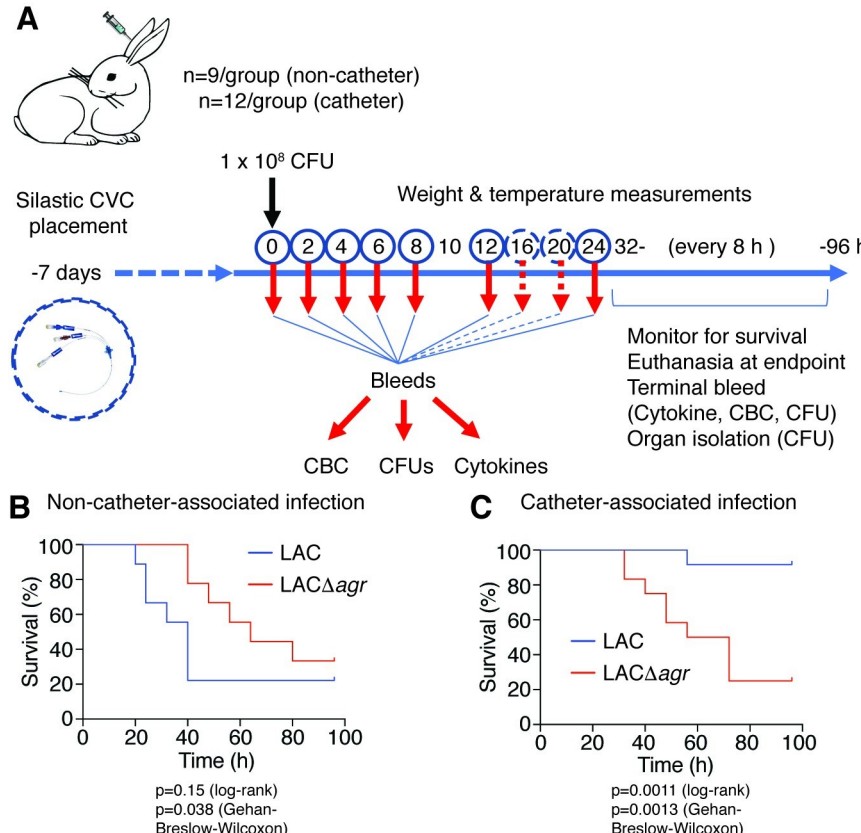

**Fig 2. Rabbit sepsis models–experimental setup and survival.** (**A**) Experimental setup of catheter- and non-catheter-associated model. The models were set up essentially in the same fashion, except that in the catheter-associated model, a silastic CVC was inserted 7 days before injection of $10^8$ CFU of *S. aureus* LAC or its isogenic Δ*agr* deletion strain and additional 16- and 20- h samples (dashed circles) were only taken in the catheter-associated model. (**B**) Survival curve in the non-catheter associated (n = 9/group) and (**C**) of the catheter-associated model (n = 12/group). The non-catheter-associated model was performed two independent times (n = 4/group and n = 5/group, respectively; total, n = 9 animal/group). The catheter-associated model was performed two independent times with n = 6 animals/group each (total, n = 12 animals/group). Data obtained with the two batches were combined in both models. Statistical analysis is by the indicated tests. Rabbit picture is from openclipart.org.

rabbit infected with the wild-type strain developed a biofilm. There were highly significant correlations between negative Agr status and death (Fisher's exact test, p = 0.0028) and between biofilm formation and death (Fisher's exact test, p = 0.0005). Together, these results show that (i) in rabbits the observed impact of Agr on the outcome of sepsis is much weaker than in mice and (ii) the effect is reversed in device-associated infection, where Agr deficiency causes higher mortality in apparent association with biofilm formation on the device.

## Early neutropenia is associated with a fatal outcome of sepsis

Leukocyte numbers decreased in the early hours of infection in the non-catheter-related infection model, while they increased again in survivors (**Figs 5A, 5B and S2**). This was most pronounced in neutrophils (**Fig 5B**), but less obvious in the catheter-related model, probably due to attachment of bacteria to the device and slow release from the formed biofilm (**Fig 5C and 5D**). We then analyzed whether early leukopenia and specifically neutropenia is correlated with outcome of infection and Agr status. Numbers of total leukocytes and all leukocyte types except monocytes were significantly lower in animals that ultimately died during early

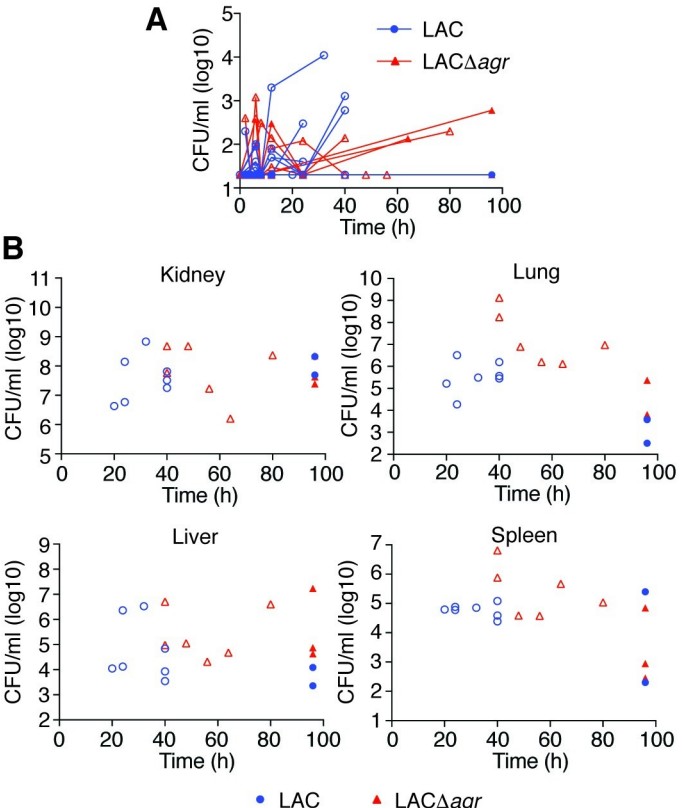

**Fig 3. Rabbit non-catheter-associated sepsis model–CFU in blood and organs.** (**A**) Bacterial CFU in the blood over time. (**B**) Bacterial CFU at the time of death or the end of the experiment in the indicated organs. Empty symbols represent animals that died during the observation period.

infection in the catheter-related model, and neutrophil numbers were significantly lower in both models (**Figs 5E–5H and S2**). The most important conclusion from this analysis is that early neutropenia is correlated with fatality in both device- and non-device associated sepsis. Notably, confirming the opposite impact of Agr status on infection outcome in catheter- versus non-catheter-related infection, total leukocyte and neutrophil numbers early during infection were significantly lower in the wild-type- as compared to Δ*agr*-infected animals in the non-catheter-related model, while they were significantly higher in the catheter-related model (**Fig 5I–5L**).

A "cytokine storm" is often regarded a hallmark of sepsis [7]. We therefore analyzed the concentration over time of specific cytokines often reported to change in concentration during sepsis. IL-6 concentrations were very strongly increased over the course of infection in both infection models and were generally higher in moribund animals than in those that recovered (**Fig 6A and 6G**). Concentrations of TNF-α followed a similar pattern but with much lower overall changes, and this was only observed in the non-catheter-associated model (**Fig 6B and 6H**). In contrast, concentrations of the anti-inflammatory cytokine IL-4 were overall reduced compared to baseline (**Fig 6C and 6I**). These findings reflect previous reports on cytokine concentrations during sepsis [7]. However, cytokine concentrations during the analyzed initial time window were not correlated with infection outcome, indicating that they are not an early predictor of infection outcome (**Fig 6D, 6E, 6F, 6J, 6K and 6L**). Consistently with the lack of correlation between cytokine concentration and death, there was also no significant correlation with Agr status (**S3 Fig**).

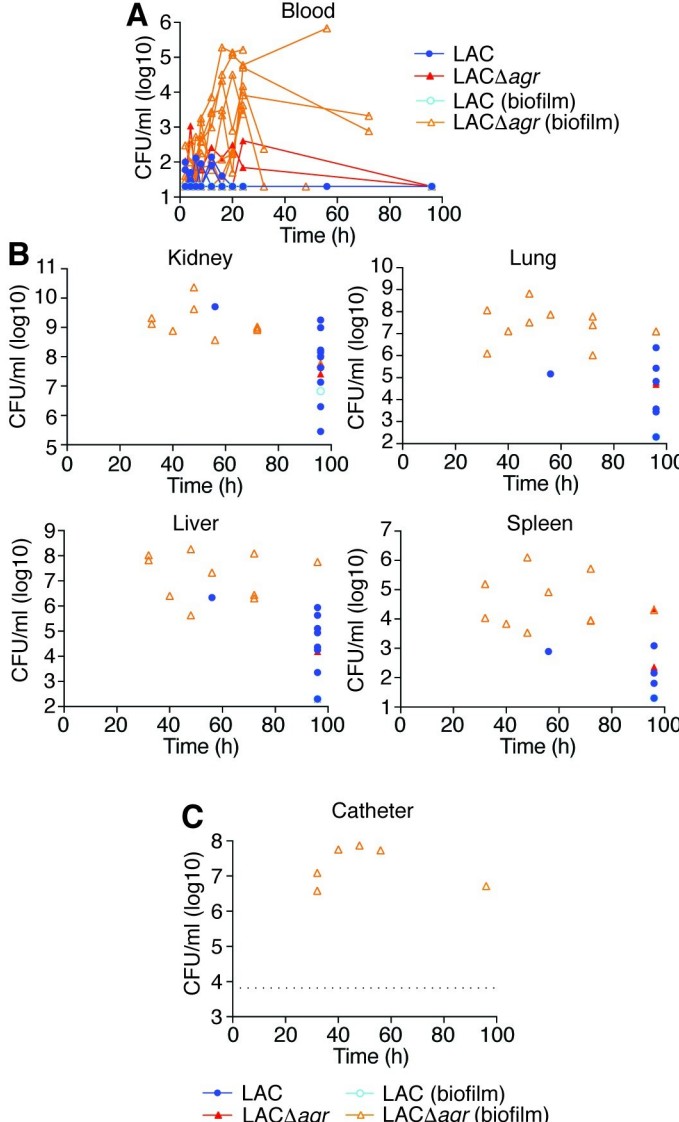

**Fig 4. Rabbit catheter-associated sepsis model–CFU in blood, organs, and on catheters.** (**A**) Bacterial CFU in the blood over time. (**B**) Bacterial CFU at the time of death or the end of the experiment in the indicated organs. (**C**) Bacterial CFU in the lumina of implanted catheters. Data are from the second batch of rabbits only (n = 6/group). The dotted line depicts the threshold above which CFU values were considered to reflect biofilm formation in both batches of rabbits. (**A-C**) Rabbits that developed catheter biofilms are marked by open symbols and light-colored borders.

## Discussion

Bacterial virulence has not traditionally been regarded as a primary factor determining the outcome of sepsis. This notion is based on conventional models that attribute sepsis to an overwhelming immune reaction to widely conserved bacterial surface structures [4–6] or more recent reports that emphasize the role of a later stage of immune suppression [8–13]. These models, which explain differences in sepsis outcome predominantly by host factors, can hardly explain the clinically observed increased morbidity and mortality that is associated with specific pathogens such as *S. aureus* [53]. In this study we hypothesized that bacterial virulence significantly affects sepsis development and outcome. To evaluate that hypothesis, we analyzed

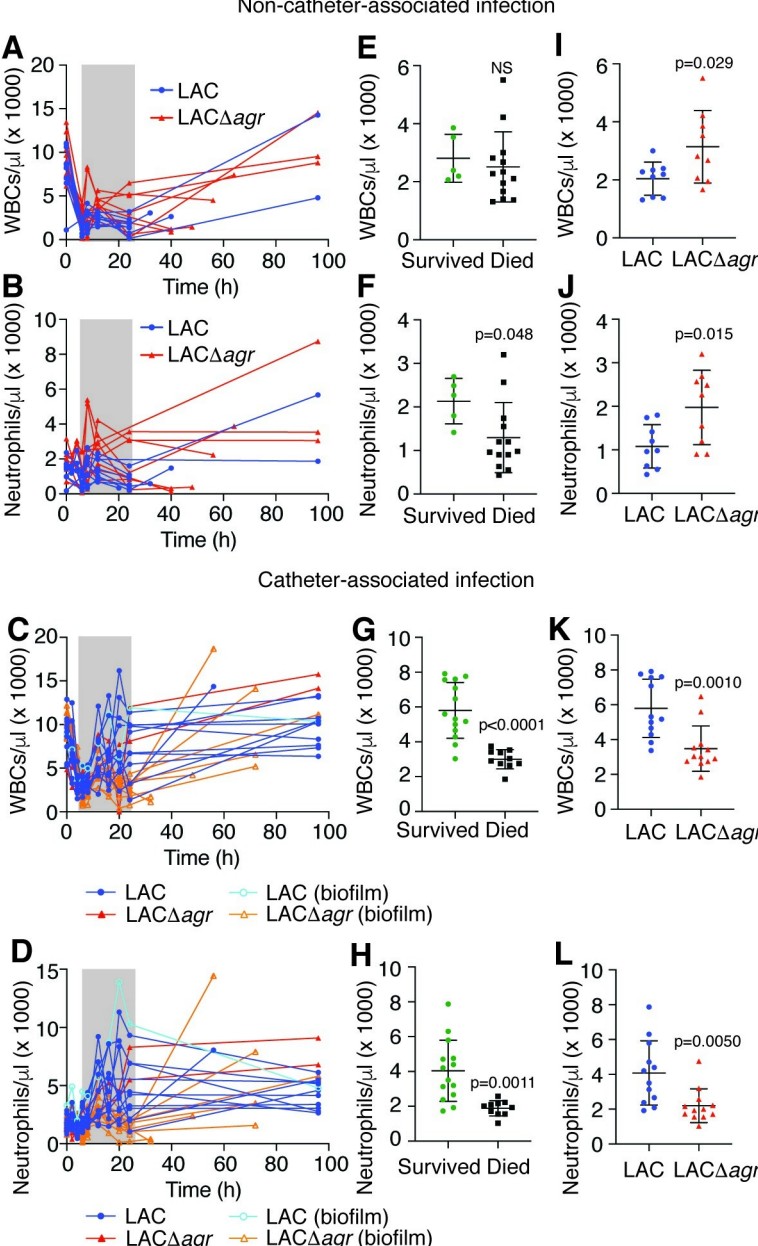

**Fig 5. Rabbit sepsis models–leukocyte numbers.** (**A-D**) WBC and neutrophil numbers over time in the non-catheter-associated (**A,B**) and catheter-associated (**C,D**) model. (**C,D**) Rabbits that developed catheter biofilms are marked by open symbols and light-colored borders. (**E-H**) Analysis of death versus survival outcome based on the averages of leukocyte numbers of every animal in the time window 6–24 h (grey shading) in panels A-D. (**I-L**) Analysis of impact of Agr status on the averages of WBC and neutrophil numbers in the time window 6–24 h (grey shading) in panels A through D. (**E-L**) Statistical analysis is by unpaired two-tailed t-tests. Error bars show the mean ± SD. NS, not significant (p≥0.05). See **S2 Fig** for monocyte and lymphocyte analyses.

the role of the global virulence regulator Agr and the association of leukopenia with mortality in acute blood infection caused by MRSA as a leading cause of sepsis. Agr and leukocyte killing are considered major parameters of *S. aureus* and MRSA virulence: Agr is a major regulator of virtually all *S. aureus* toxins, among them many leukocidins [29,52], and while only directly

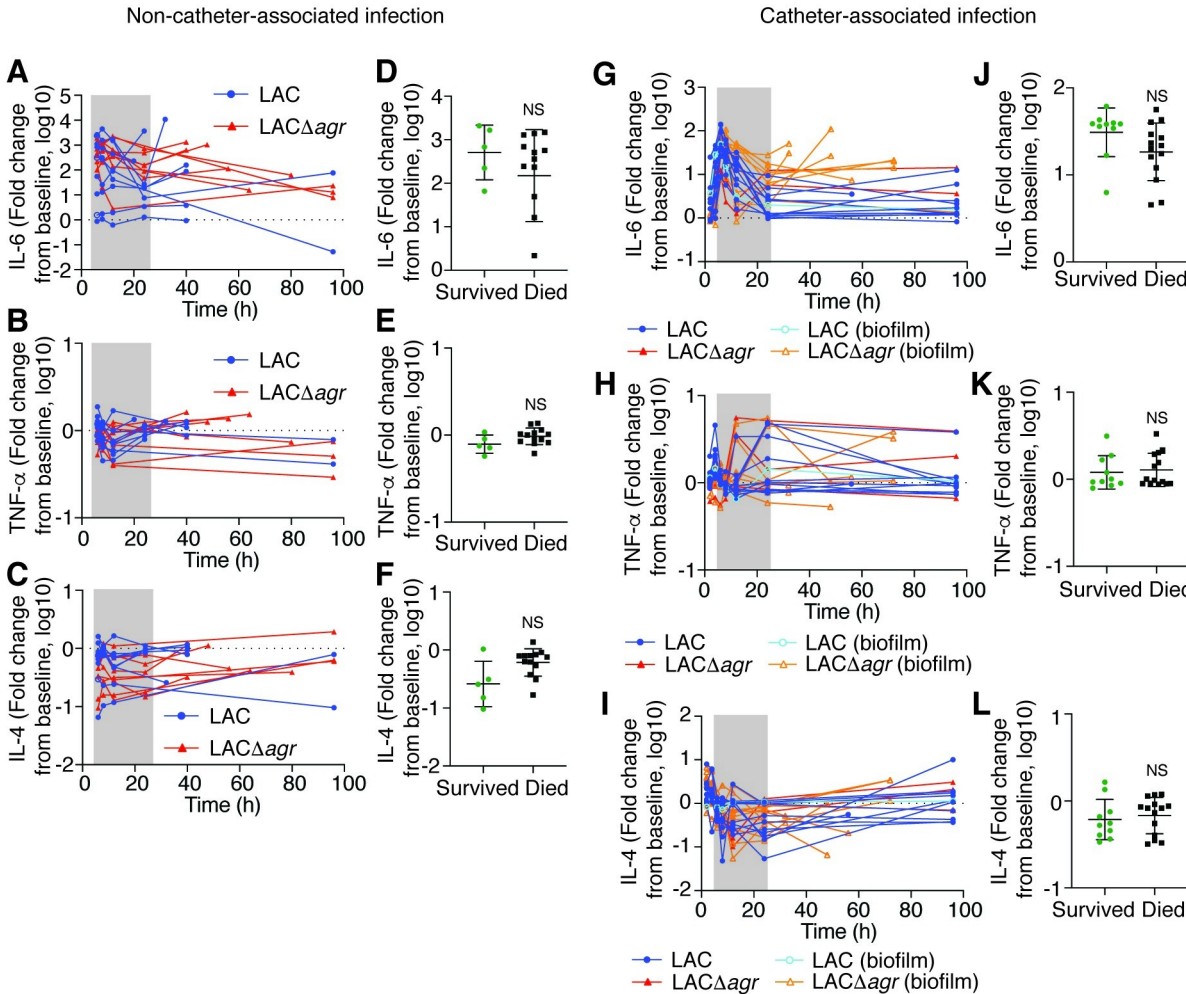

**Fig 6. Rabbit sepsis model–cytokine levels.** (**A**-**C**) Fold-changes of the indicated cytokines over time in the non-catheter-associated model. (**D**-**F**) Analysis of death versus survival outcome based on average cytokine fold-changes in the time window 6–24 h (grey shading) in panels A-C. (**G**-**I**) Fold-changes of the indicated cytokines over time in the catheter-associated model. (**J**-**L**) Analysis of death versus survival outcome based on average cytokine fold-changes in the time window 6–24 h (grey shading) in panels G-I. (**D**-**F**, **J**-**L**) Statistical analysis is by unpaired two-tailed t-tests. Error bars show the geometric mean and geometric SD. NS, not significant (p≥0.05). See **S3 Fig** for the corresponding analysis of Agr status for both rabbit models.

demonstrated in a pneumonia model [54], leukopenia is the assumed in-vivo consequence of their activity. We chose these parameters as they allow for a better general evaluation of our hypothesis than the analysis of specific toxins, given the plethora of *S. aureus* toxins and their functional redundancy [55]. Our findings on mortality being associated with the virulence regulator Agr in addition to the absence of any effect under CY treatment indicate a significant impact of bacterial virulence on the outcome of sepsis that occurs via leukocyte killing. This notion is also supported by the kinetics of the observed leukopenia, because overall leukocyte and neutrophil numbers first increased, but then rapidly decreased, strongly suggesting that low leukocyte concentrations in moribund animals were a result of lysis rather than leukocyte migration into organs, which happens much earlier, at ~ 5 min post infection [56]. This migration-dependent leukopenia is short-timed and leukocyte numbers return to normal at 60 min, which is long before the effects we measured [56].

Our study revealed differences in mouse versus rabbit test animals regarding the relative impact of Agr during non-catheter-associated infection, inasmuch as the relative impact of Agr was stronger in mice. We believe this to be due to the fact that mice are immune to many leukocidins owing to the lack of responsive receptors [47]. While rabbits are therefore overall more sensitive to infection by *S. aureus*, the relative impact of Agr–which increases with the relative Agr dependence of toxin expression–is greater in mice, as in that species it is mostly dependent on PSMs, which are under much stricter Agr control than the bicomponent leukocidins [40,52]. As rabbits more closely resemble humans in their responsiveness to leukocidins, this finding indicates that the impact of Agr on systemic infection may be overestimated in the mouse infection model. Furthermore, we are aware of the fact that these toxins not only have cytolytic, but also pro-inflammatory capacities, often detected at concentrations much lower than those that are cytolytic [57,58]. Moreover, some toxins, such as PSMs and LukAB/GH, are assumed to contribute to leukocyte lysis after phagocytosis rather than by toxin release into the bloodstream [59,60]. Which specific leucocidin activities contribute to pathogenesis, including during different experimental setups that may differ for example in the initial bacterial inoculum, is not understood.

While our study was focused on assessing the role of bacterial virulence, it gave important information also regarding immune regulation in sepsis. Notably, in our animal studies we analyzed very early stages of sepsis, which is not commonly possible in the clinic. We found that early leukopenia (particularly neutropenia), but not early increase of pro-inflammatory cytokines, was strongly correlated with mortality. Our results indicate that immune suppression early during infection plays a critical role for the development of sepsis. This highlights the importance of different stages of sepsis regarding inflammation and immune suppression. Of note, the early immune suppression we observed, which is mediated by bacterial virulence, is different in character and timing from the previously described late immune suppression that is considered a host effect [8–13].

While many animal models of device-associated infection have been performed, particularly using *S. aureus* as pathogen [28,41,42,61–64], we here also performed what to the best of our knowledge represents the first experimental assessment of device-associated sepsis. As mentioned above, this is of particular importance given that most cases of staphylococcal sepsis originate from the infection of indwelling medical devices, which due to the constant seeding from the bacterial biofilm on the infected device represents a completely different scenario [26,27]. Results of the device-associated infection model generally reflected those of the non-device-associated model as for bacteria-leukocyte interaction, but were opposite regarding the role of Agr. Due to complete absence of the PSMs, which are important for biofilm structuring and turnover [39,42,65], Agr-deficient *S. aureus* are known to form extended biofilms in vitro [43] and on indwelling medical devices in vivo [41,42]. However, it has not been previously investigated how this situation impacts the development of sepsis as the most severe complication of device-associated infection. Our results demonstrated increased mortality, accompanied by increased bacterial presence in blood and organs, in rabbits infected with Agr-deficient as compared to those infected with wild-type MRSA, effects that were more pronounced in animals in which biofilms had formed on the device. Only one animal infected with wild-type bacteria developed a biofilm on the device, but did not show comparably severe disease parameters, in accordance with the notion that in-vivo biofilms formed by wild-type *S. aureus* are less extended and less resistant to removal by phagocytes than those formed by Agr-deficient strains [41]. These results show that Agr dysfunction and concomitant biofilm formation on the indwelling device causes increased development of sepsis, most likely caused by constant seeding of bacteria from the device. Notably, they help to explain the clinical

observation of a high percentage of functionally Agr-deficient isolates found in cases of *S. aureus* blood infection [36].

Our results have important implications for the suggested application of quorum-sensing blockers for the treatment of *S. aureus* systemic infection. Such drugs have often been proposed and reported to have efficacy in local and occasionally systemic *S. aureus* infections [66–68]. An inherent problem with the interpretation of results achieved with quorum-sensing blockers is the frequently narrow window between genuine quorum-sensing blocking efficacy and bactericidal effects. The comparison of wild-type with isogenic Δ*agr S. aureus* thus gives a better idea than the use of quorum-sensing blocking drugs about whether Agr is a promising target. The frequent involvement of devices in *S. aureus* bacteremia [26,27], the fact that clinical treatment even of sepsis without a device origin often requires placement of devices such as catheters, and the increased pathogenesis associated with Δ*agr S. aureus* in device-associated systemic infection that we found argue against using quorum-sensing blockers for systemic *S. aureus* infections. Because strict control of PSMs by Agr is the underlying cause for these phenomena [40,42,65], specifically targeting PSMs also does not appear to be a promising drug development strategy for blood infections. Our results on the importance of leukocyte-bacteria interaction for the outcome of sepsis rather suggest that protection of leukocytes from bacterial attacks, such as by direct interference with non-PSM bacterial leukocidins [69–71], bears promise for the treatment of device- or non-device-associated *S. aureus* blood infection.

In conclusion, in this study we provide evidence for a major contribution of bacterial virulence to sepsis and show that early neutropenia is a predictor for sepsis outcome. These results call for an adjustment of the traditional notion that the severity of bacterial sepsis is predominantly attributable to host factors. In addition, our results give important mechanistic information regarding device-associated staphylococcal sepsis to guide the development of alternative anti-virulence strategies.

## Materials and methods

### Ethics statement

Mouse and rabbit experiments were performed under NIAID animal study protocols LB1E and LB2E, respectively. Both protocols were reviewed and approved by the Animal Care and Use Committee (NIAID, NIH) and are in accordance with the Animal Welfare Act of the United States (7 U.S.C. 2131 et. seq.).

### Bacterial strains and growth conditions

*S. aureus* strains used in this study were LAC (CA-MRSA, pulsed-field type USA300) [72] and its isogenic *agr* mutant [57]. Bacteria were prepared for infections as described previously [20].

### Mouse model of non-catheter-associated blood infection

The mouse sepsis model was performed using female, 8–10 week-old, C57BL/6NCrl mice (Charles River Laboratories) as described previously [73] using tail vein injection with the following modifications. Mice received ~ $2-3 \times 10^8$ CFUs of live *S. aureus*. The health status of challenged mice was recorded every 2 hours for the first 12 hours, every 4 hours for the following 12 hours, and every 8 hours thereafter until the study ended (52 hours). Mice that prematurely reached the end point (defined as having a head tilt, microphthalmia, lack of response to prodding, immobility, or >20% weight loss from the pre-infection baseline) during the study and those that survived to the end of the study were anesthetized with isoflurane. For depletion of leukocytes in naïve mice, cyclophosphamide monohydrate (CY, Sigma), diluted in sterile

PBS, was injected intraperitoneally at 250 mg/kg each day for 3 days prior to intravenous infection with $1 \times 10^6$ CFUs of bacteria on day 4. This lower dose of bacteria was chosen because of generally increased sensitivity to infection of CY-treated animals. CBC analyses of blood from CY-treated mice confirmed significantly reduced numbers of WBCs (~ 20/ml) compared to untreated mice (~8000/ml). The health status of challenged mice was recorded as described above until the study ended.

## Rabbit models of systemic catheter- and non-catheter-associated blood infection

Female, 2–3 kg, New Zealand White rabbits (Charles River Laboratories) were housed singly and allowed to acclimatize in an American Association for the Accreditation of Laboratory Animal Care (AAALAC)–approved NIH facility for a minimum of 7 days prior to introduction of catheters and/or infection with *S. aureus*.

Vascular catheterization was performed using in-house made sterilized central venous catheters (CVCs), composed of silastic catheters (1.02 × 2.16 mm, Dow Corning) attached to 18-gauge intramedic Luer stub adapters (Becton Dickenson). CVCs were sterilized with ethylene oxide gas and then cut to length for each rabbit. Hair, over the right cervical, right shoulder and scapular region for each rabbit was removed. Under anesthesia, a 2- to 3-cm-long skin incision was made on the right anterolateral cervical region to expose the external jugular vein. A small incision was then made through the vein wall and a sterile-saline filled CVC was threaded into the vein and forwarded until the catheter cuff was contiguous with the vein. The proximal end of the CVC was exteriorized by a subcutaneous tunnel created by threading it through a sterile trocar (1 × 34 cm) underneath the skin from the external jugular incision site to an exit site through the intrascapular skin. The skin incision over the external jugular vein was then closed and sutured. The proximal part of the catheter was attached to the metal end of the Luer stub and a needleless injection cap, which was secured to the skin exit with 2–0 silk sutures and tied together to prevent displacement [74]. Two-way flow was performed to confirm integrity of the catheter with up to 1 ml of sterile pharmaceutical-grade heparin in physiological saline solution (100 units/ml). Catheterized rabbits were placed on post-operative observation for 7 days post-surgery prior to intravenous infection with *S. aureus*.

For pre-infection baselines, weights and rectal temperatures were taken prior to infection. Additionally. blood samples were drawn from the ear artery or via the catheter of non-catheterized and catheterized rabbits, respectively. Blood samples were drawn from the ear artery and weights and rectal temperatures were taken prior to infection for pre-infection baselines. Rabbits (with and without catheters) were then infected via the marginal ear vein with 1 ml of bacteria at a concentration of $1 \times 10^8$ CFUs/ml in sterile PBS. Approximately 0.5-ml volumes of blood were sampled 2, 4, 6, 8, 12, and 24 hours post infection from the marginal ear vein of non-catheterized rabbits, and 2, 4, 6, 8, 12, 16, 20 and 24 hours post infection via the catheter of catheterized rabbits, for quantitative blood culture, cytokine measurements and CBC analyses. The health status, weights and rectal temperatures were recorded every two hours post infection for the first 8 hours, every 4 hours for the following 16 hours, and then every 8 hours thereafter until the end of the study (96 hours). Rabbits prematurely reaching end-point (defined as having neurological signs, abdominal breathing, cyanosis, a lack of response to prodding, immobility, or >15% weight loss from the pre-infection baseline for two consecutive days) during the study and those that survived to the end of the study, were anesthetized with ketamine and xylazine and terminally exsanguinated. Rabbits were then euthanized with Beuthanasia D (Schering-Plough Animal Health Corp) and the livers, kidneys, spleens, lungs and any catheters were surgically removed. Enumeration of CFUs from organ homogenates

was performed as previously described [75]. The catheter lumina were flushed several times with a total of 0.5 ml of sterile PBS each, homogenized, and serial dilutions of the contents were plated onto TSA plates for enumeration of CFU. Biofilm formation was defined as a result of > 7500 CFU/ml. No biofilm formation was defined as a result of < 200 CFU/ml (detection limit). There were no catheters with CFU between these two values.

## CFU enumeration, cytokine measurements and complete blood count analyses from blood

Blood samples were distributed into tubes containing serum clotting gel, heparin, and ethyl-enediaminetetraacetic acid (EDTA) (Sarstedt) and then inverted immediately. The total volume of blood draws did not exceed those determined by Comparative Medicine Branch Standard Operating Protocol guidelines. The concentrations of IL-4, IL-6, and TNF-α in rabbit sera (stored at -20˚C immediately after collection) were determined using Quantikine ELISA kits (R&D Systems) according to the manufacturer's instructions. CFUs were enumerated from blood samples collected in heparin-containing tubes as described previously (20). CBCs were assessed in blood samples collected in EDTA-containing tubes within 3 hours of collection using a ProCyte Dx Hematology Analyzer (IDEXX technologies). Automatic gating, defined by the software algorithms, was used to identify white blood cell differentials on all uninfected samples. For blood samples collected from infected animals, manual gating was applied to identify white blood cell differentials. Identical gates were applied across samples collected at each time point.

## Statistics

Statistical analysis was performed using Graph Pad Prism for MAC version 8.3.0. Statistical analysis is by Log-rank (Mantel-Cox) and/or Gehan-Breslow-Wilcoxon tests for survival curves. For the comparison of two groups, unpaired two-tailed t-tests were used. All error bars depict the mean and standard deviation for non-logarithmic, or the geometric mean and geometric standard deviation for logarithmic scales.

## Supporting information

**S1 Fig. Physiological parameters in the rabbit sepsis models.** Weight change and rectal temperature over time in the non-catheter-associated and the catheter-associated sepsis models. The transient dip at ~ 48 h in weight for some animals in the catheter-associated model was due to those rabbits not eating. After critical care (forced feeding) was activated, they recovered.
(TIF)

**S2 Fig. Rabbit sepsis models–leukocyte numbers (monocytes, lymphocytes).** (**A-D**) Monocyte and lymphocyte numbers over time in the non-catheter-associated (**A,B**) and catheter-associated (**C,D**) model. (**C,D**) Rabbits that developed catheter biofilms are marked by open symbols and light-colored borders. (**E-H**) Analysis of death versus survival outcome based on average monocyte and lymphocyte numbers for every animal in the time window 6–24 h (grey shading) in panels A-D. (**I-L**) Analysis of impact of Agr status on average monocyte and lymphocyte numbers for every animal in the time window 6–24 h (grey shading) in panels A through D. (**C-L**) Statistical analysis is by unpaired two-tailed t-tests. Error bars show the mean ± SD. NS, not significant (p≥0.05).
(TIF)

**S3 Fig. Impact of Agr status on cytokine levels early during infection.** Fold-changes in average cytokine levels for every animal in the rabbit infection models in the early 6–24 h infection window were analyzed dependent on infection group (wild-type versus Δ*agr*-infected animals). Statistical analysis is by unpaired two-tailed t-tests. Error bars show the geometric mean and geometric SD. NS, not significant (p≥0.05).
(TIF)

## Acknowledgments

We would like to thank Dr. Binh A. Diep (UCSF, CA) for providing advice on the rabbit experiments, Dr. Marvin L. Thomas, Dr. Holly Habbershon, Mr. Keith Johnson, and Mr. John DeLeonardis at the Division of Veterinary Resources (DVR), NIH, for the surgical placement of catheters in rabbits and their invaluable technical expertise, and the technical staff at the DVR and Comparative Medicine Branch (CMB), NIH, for excellent technical support.

## Author Contributions

**Conceptualization:** Gordon Y. C. Cheung, Michael Otto.

**Data curation:** Gordon Y. C. Cheung, Justin S. Bae, Michael Otto.

**Formal analysis:** Gordon Y. C. Cheung, Justin S. Bae, Michael Otto.

**Funding acquisition:** Michael Otto.

**Investigation:** Gordon Y. C. Cheung, Justin S. Bae, Ryan Liu, Rachelle L. Hunt, Yue Zheng.

**Methodology:** Gordon Y. C. Cheung, Michael Otto.

**Project administration:** Gordon Y. C. Cheung, Michael Otto.

**Resources:** Michael Otto.

**Supervision:** Gordon Y. C. Cheung, Michael Otto.

**Validation:** Gordon Y. C. Cheung, Justin S. Bae, Michael Otto.

**Visualization:** Gordon Y. C. Cheung, Justin S. Bae, Michael Otto.

**Writing – original draft:** Gordon Y. C. Cheung, Michael Otto.

**Writing – review & editing:** Gordon Y. C. Cheung, Justin S. Bae, Michael Otto.

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
