## [Decision Letter · Decision Letter 0]

5 Jun 2020

Dear Dr. Otto,

Thank you very much for submitting your manuscript "Bacterial virulence plays a crucial role in MRSA sepsis" for consideration at PLOS Pathogens. As with all papers reviewed by the journal, your manuscript was reviewed by members of the editorial board and by several independent reviewers. In light of the reviews (below this email), we would like to invite the resubmission of a significantly-revised version that takes into account the reviewers' comments.

As you will see from the reviews, all referees thought the work has potentially important implications. However, all three were concerned about the relatively sparse data to support the major conclusions of the paper. In particular, R1 and R2 would like to see more thorough investigation of the catheter biofilms. All reviewers noted that the correlation between leukopenia and mortality was not strongly supported by the data, and R1 points out that the mechanism of leukopenia is not investigated. R3 raises important questions about the robustness of the statistical analysis and the hazards of drawing conclusions on the basis of apparently a single experiment with a small number of animals. Confirming the results of the rabbit experiment in another group of animals would greatly strengthen the conclusions.

We cannot make any decision about publication until we have seen the revised manuscript and your response to the reviewers' comments. Your revised manuscript is also likely to be sent to reviewers for further evaluation.

Sincerely,

Michael R. Wessels

Section Editor

PLOS Pathogens

Kasturi Haldar

Editor-in-Chief

PLOS Pathogens

orcid.org/0000-0001-5065-158X

Michael Malim

Editor-in-Chief

PLOS Pathogens

orcid.org/0000-0002-7699-2064

As you will see from the reviews, all referees thought the work has potentially important implications. However, all three were concerned about the relatively sparse data to support the major conclusions of the paper. In particular, R1 and R2 would like to see more thorough investigation of the catheter biofilms. All reviewers noted that the correlation between leukopenia and mortality was not strongly supported by the data, and R1 points out that the mechanism of leukopenia is not investigated. R3 raises important questions about the robustness of the statistical analysis and the hazards of drawing conclusions on the basis of apparently a single experiment with a small number of animals. Confirming the results of the rabbit experiment in another group of animals would greatly strengthen the conclusions.

Reviewer's Responses to Questions

**Part I - Summary**

Reviewer #1: Cheung and colleagues studied staphylococcal pathogenesis in models of catheter and non-catheter related sepsis both in mice and rabbit. To follow virulence functions regulated by agr, particularly toxins, they compared infection induced by WT and agr mutants. Overall, rabbits nicely modeled human sepsis and showed several clinical and lab features that approximated what is seen in human sepsis. For example, like in human infection, low WBC was shown to be a prognosticator of poor outcome, and agr negative S. aureus is associated with Staphylococcal bacteremia. The authors proposed that lysis of WBC by toxins accounted for the low WBC and drives poor outcome, although that needed better support. Finally, because negative agr is associated with worse outcome in catheter infection, the authors caution on the use of quorum sensing blockers for treatment of S. aureus sepsis. Overall, the study has potentially important clinical implications, but mechanistic aspects could be improved.

Reviewer #2: This paper examines the contribution of virulence factors during S. aureus sepsis.

The authors examine the possibility that sepsis is not only the result of excessive inflammation. The authors surmise that bacterial factors also contribute to sepsis. They hypothesize that secreted leukocidins may actively kill leukocytes early during infection exacerbating sepsis. Because of the redundancy of secreted leukocidins, the authors settle into comparing the infectivity of the wild type LAC strain and the isogenic mutant in the global regulator Agr which controls the expression of many leukocidin genes amongst many other genes. In addition, the authors use both a mouse and rabbit models of IV infection with and without catheter induced bacteremia. This is done because mouse leukocytes are in general resistant to S. aureus leukocidins unlike rabbit leukocytes.

The study addresses an important question in the field using a rational design. But, the interpretation of the results is constrained by the small number of data points.

Reviewer #3: Authors present findings of animal models of infection with WT MRSA and an isogenic agr-deletion strain to demonstrate the role of agr-mediated quorum-sensing in the pathogenesis and subsequent outcome of sepsis. Major conclusions are that bacterial leukocidins up-regulated via agr-mediated mechanisms during early sepsis contribute to leuko- and neutropenia that drives fatal outcomes. In addition, a model of IV infection in rabbits with a pre-inserted central venous catheter (CVC) surprisingly shows that agr-deficiency is associated with higher mortality that is associated with biofilm formation. Taken together, the implications of such findings could be highly significant, as they would suggest that treatment modalities targeting S. aureus virulence factors could effectively improve staphylococcal sepsis outcomes, but that targeting quorum-sensing may be the wrong approach in device- or biofilm-associated infection. The experiments were well-executed by researchers with deep experience in the methods and the field, although small numbers of animals, particularly in the catheter-associated infection model, hamper generation of statistically-supported conclusions in several experiments. Such high impact conclusions should be substantiated by more appropriate statistical analyses in some experiments, and this may not be feasible with the numbers of animals used in the experiments represented.

Authors describe two major experiments from which they ultimately draw their conclusions. First, using a mouse model of MRSA sepsis from IV infection with a WT (1 of 10 survived) or an isogenic agr-deletion LAC strain (all survived), they demonstrate the critical role of agr-mediated quorum-sensing in mortality. From blood draws early after infection, they show that survival was associated with higher leukocyte and leukocyte subset numbers. They repeat this experiment with cyclophosphamide administered prior to infection to show that leukocyte-depletion renders infection even with the agr-deletion strain fatal. Taken together, they conclude that leukopenia is associated with fatal outcome in this model, and that agr-regulated expression of leukocidins drives this critical deficit.

In their second experiment, they use a similar infection model with the same MRSA strains in rabbits with and without prior placement of a CVC. In the non-catheter-related infection, mortality was similar but delayed following infection with the agr-deletion strain (3 of 9 survived versus 2 of 9 survivors from WT infection). Surprisingly, in rabbits with a CVC, 6 of 6 animals survived IV infection with the WT strain while only 2 of 6 animals survived IV infection with the agr-deletion strain, and the 4 rabbits who succumbed had biofilm detected in their catheter by CFUs counted from postmortem catheter flushes (while the 2 survivors in this group had no biofilm). Survivors in both the non-catheter- and catheter-related models had higher neutrophil counts than non-survivors. Cytokine evaluation early after infection revealed elevated IL-6 amongst all animals with no difference among survivors and non-survivors but minimal change and no difference in TNFa levels. Taken together, authors conclude that (1) the impact of agr-mediated pathophysiology in non-catheter-related sepsis is weaker in rabbits than in mice and (2) the agr-mediated effect is reversed in catheter-associated sepsis in rabbits and this reversal is due to biofilm formation and (3) early leuko- and neutropenia is associated with mortality in non-catheter and catheter-related infection while levels of canonical inflammatory-associated cytokines were not.

**Part II – Major Issues: Key Experiments Required for Acceptance**

Reviewer #1: 1) Reduced WBC as a prognosticator of poor outcome (a major point of this paper) is ascribed to toxin mediated cytolytic activity, but no supporting data is provided. The authors referred to a SA pneumonia paper where cytolysis is demonstrated. This needs to be better supported in the bloodstream infection model.

2) agr mutant caused greater mortality than WT S. aureus in the biofilm model. It makes sense that this could be secondary to slow and prolonged release from the catheter. Did the authors culture the catheter directly to demonstrate improved binding to the catheter, rather than indirectly through blood cultures, to correlate with the higher concentration in the blood?

Ideally blocking of biofilm related function within agr would lead to similar outcome as delta agr. At least discuss.

Reviewer #2: 1.Fig. 1A and 2B: How do the authors account for the difference in mortality associated with the agr mutant between mice and rabbits? The results seem counter-intuitive given the notion that the mouse should be insensitive to leukocidins controlled by Agr (as pointed by the authors).

2.l. 171-173 (Fig. 4): the authors state that 4 of 6 rabbits infected with the agr strain developed biofilms on the device and all these animals succumbed to infection. It would have been nice to report CFUs in biofilms or at least images of LAC and agr biofilms. Presumably some of the LAC infected animals also developed biofilm (Fig. 4A). It is hard to see the blue lines in this graph. Does this mean that biofilm in LAC infected animals do not release bacteria in the bloodstream?

3.Fig. 5: The authors conclude that neutropenia correlates with fatality in both device- and non-device associated sepsis (lines 186-187). However, when looking at Fig. 5D and Fig. 5H, it is hard to make such a conclusion. In panel 5D, animals survived with an average neutrophil count of 2000 per microliter. But in panel 5H, it seems that for the same average, all the animals died; rather 5000 neutrophils per microliter were required for survival.

4.The authors compare blood cell counts between surviving and dead animals throughout the manuscript. Do the authors observed any sort of cell count correlation when comparing the 2 strains used for infection?

Reviewer #3: 1.The data authors cite to justify the conclusion that leukopenia/neutropenia early after infection is associated with morbidity/death are not convincing as shown. If interpreting correctly from their methods description, they included repeated measures from the same 5-6 animals across several 2-hour intervals, but grouped all of these results stratified by survivors and non-survivors for comparison. That seems akin to having several technical replicates (repeated measures from the same animal) represent biological replicates for this critical comparison. This analysis should be more accurately represented, perhaps by measuring the fold-change in cell counts during the first 24 hours in each animal, or using an individual and consistent timepoint for each animal (ie- last CBC prior to moribund or last available for all animals) and comparing these between the groups.

2.In general, many of the conclusions inferred from the data in regards to recovered CFUs, WBC and neutrophil trends, and cytokine findings are based on trends without applied statistics (some examples are specifically cited as Minor Issues). This likely reflects lack of power to appropriately apply statistical comparisons between groups due to small numbers, but it generally weakens the strength of several of the major conclusions drawn by the authors.

3.The finding that rabbits with a CVC infected with the WT MRSA strain all survived the infection is very surprising and contrasts directly with the findings from the non-catheter-associated infection, yet there is no description from the authors to explain this finding. It is counter to the conclusion that agr-mediated leukocidin expression causes leuko- and neutropenia that leads to mortality and is not well explained by the possibility that bacteria seed the catheter early and are thus sequestered there since the leukocidins would be expected to be secreted. Likewise, explanation of the decreased impact of agr-deficiency on outcome in the non-catheter-associated infection in rabbits compared to mice is lacking.

**Part III – Minor Issues: Editorial and Data Presentation Modifications**

Reviewer #1: 3) While cytolysis is a phenomenon that is readily observed in animals particularly when high inoculums of SA is injected, it is unclear how relevant that is with lower inoculum infections such as bloodstream infection (in humans). Toxins have important pro-inflammatory and cytolytic activity at different concentrations. This could be further discussed.

4) Line 204 – “Sepsis has not traditionally been thought to be impacted significantly by bacterial

205 virulence” – Not clear if this is a general consensus. Please provide a reference

5) Figure 3 and 4 - CFUs collected at time of death are not particularly useful. Probably better in supplement.

Reviewer #2: 1.Fig. 1B: Data was not collected at time 0 to ascertain the range of WBCs (neutrophils, monocytes, lymphocytes). The authors assume that all animals add the same number of neutrophils at time 0 but is this truly the case?

2.Fig. 2: Were bacterial inocula prepared similarly for infections shown in panels B and C?

3.The method regarding mouse infection (l. 302-303) should describe the preparation of S. aureus bacteria. Reference 20 describes a paper for S. epidermidis. It is okay to be specific here and confirm that authors performed tail vein injection.

4.l. 163: replace Dagr with “delta”agr.

5.Some of the sentences read awkwardly throughout the text especially in the Abstract and Author Summary.

Reviewer #3: 1.In Figure S1, p-values are described in the figure legend and referenced in the text but are not shown in the images.

2.In Figure 3B, could moribund animals be denoted with empty circles as in Fig S1? It would help clarify the statement that “CFU over the time course of infection in the blood and organs reflected the mortality results, inasmuch as they were (i) increased in moribund animals, at least in the blood and some organs” in lines 165-167.

3.The statement that “Leukocyte and neutrophil numbers decreased in the early hours of infection in the non-catheter-related infection model, while they increased again in survivors (Fig. 5A,B,E,F). This was not observed in the catheter-related model” in lines 179-181 is not clearly supported by the data shown.

4.The statement that “Concentrations of TNF-� followed a similar pattern but with much lower overall changes (Fig. 6B,H).” in lines 192-193 is not corroborated by the data shown, especially in the non-catheter-associated infection.

5.The statement that “However, we observed generally increased concentrations of pro- and decreased concentrations of anti-inflammatory cytokines over the course of the infection. Our results indicate that immune suppression early during infection plays a critical role for the development of sepsis, while later inflammatory phenomena appear associated with a moribund stage” in lines 231-235 are not corroborated by the data shown. In particular, the distinction of later inflammatory phenomena being associated with moribund stage are not supported by either the kinetics as shown or the lack of difference detected in survivors versus non-survivors.

PLOS authors have the option to publish the peer review history of their article (what does this mean?). If published, this will include your full peer review and any attached files.

Reviewer #1: No

Reviewer #2: No

Reviewer #3: No
---

## [Decision Letter · Decision Letter 1]

1 Feb 2021

Dear Michael,

Thank you very much for submitting your manuscript "Bacterial virulence plays a crucial role in MRSA sepsis" for consideration at PLOS Pathogens. As with all papers reviewed by the journal, your manuscript was reviewed by members of the editorial board and by independent reviewers. The reviewers appreciated the attention to an important topic. Based on the reviews, we are likely to accept this manuscript for publication, providing that you modify the manuscript according to the review recommendations.

The reviewers felt that the revised manuscript is substantially improved with addition of additional animal data and revisions to the text.  However, I must agree with Reviewer 3 that a pervasive problem remains with the statistical analysis of the animal challenge experiments.  Unless the reviewer and I misunderstand the methodology, multiple data points from an individual animal are treated as independent values, an approach which inflates the statistical power of comparisons between groups.  I urge you to consult with a statistician to reanalyze these data using an appropriate method for repeated measures or another approach to define a single representative value for a particular time window for each animal.  I understand that such an analysis may not yield statistical significance for every comparison, but I feel it is essential to use a rigorous statistical methodology.

Sincerely,

Mike

Michael Wessels

Section Editor

PLOS Pathogens

Kasturi Haldar

Editor-in-Chief

PLOS Pathogens

orcid.org/0000-0001-5065-158X

Michael Malim

Editor-in-Chief

PLOS Pathogens

orcid.org/0000-0002-7699-2064

Reviewer Comments (if any, and for reference):

Reviewer's Responses to Questions

**Part I - Summary**

Reviewer #1: The manuscript presents conceptual important findings.

Although the role of toxins has not been fully corroborated, the authors have satisfactorily addressed my concerns.

Reviewer #3: The authors’ revision addresses many of the questions and concerns raised in review of the original manuscript. Namely, the increased number of animals in the critical catheter-related infection model helps to strengthen many of the conclusions drawn about the differential role of agr-mediated effect on virulence in this model. The additional statistical analyses facilitated by these numbers (such as the correlations between agr status or biofilm formation and mortality) also bolster the strength of these conclusions. And the enhanced description of possible reasons for the differences in the role of agr in mice versus rabbits and in catheter- versus non-catheter-related infections substantiate these findings and provide a clearer understanding of what knowledge gaps remain to more fully understand the mechanisms of pathogenicity mediated by agr-regulated toxins.

**Part II – Major Issues: Key Experiments Required for Acceptance**

Reviewer #1: (No Response)

Reviewer #3: 1. In reference to the following comment from the original review: “The data authors cite to justify the conclusion that leukopenia/neutropenia early after infection is associated with morbidity/death are not convincing as shown. If interpreting correctly from their methods description, they included repeated measures from the same 5-6 animals across several 2-hour intervals, but grouped all of these results stratified by survivors and non-survivors for comparison. That seems akin to having several technical replicates (repeated measures from the same animal) represent biological replicates for this critical comparison. This analysis should be more accurately represented, perhaps by measuring the fold-change in cell counts during the first 24 hours in each animal, or using an individual and consistent timepoint for each animal (ie- last CBC prior to moribund or last available for all animals) and comparing these between the groups.” Authors’ cite the many challenges in comparing a similar timepoint or a standardized fold-change given difficulty in acquiring a complete set of data due to blood draw limits and technical feasibility in the rabbit experiments. While these challenges can certainly be appreciated, the chosen representation combining multiple timepoints from the same animal in the comparator groups (survived versus died or WT versus delta-agr) may be misleading. Since these data are cited to support one of the major findings from this work, that leukopenia early after infection predicts mortality from S. aureus sepsis, this analysis method should be reconsidered or should perhaps be cited as one possible limitation of the findings as presented.

**Part III – Minor Issues: Editorial and Data Presentation Modifications**

Reviewer #1: (No Response)

Reviewer #3: 1. The light blue line and open triangle used to denote LAC (biofilm) is too light. This symbol seems to only account for one animal, but it is hard to see in any of the panels.

2. It seems in Figure 3, the intent is to denote animals that survived with closed symbols and those that did not with open, but this should be clarified either on the graphs or in the legend for Figure 3.

PLOS authors have the option to publish the peer review history of their article (what does this mean?). If published, this will include your full peer review and any attached files.

Reviewer #1: No

Reviewer #3: No
---

## [Editor Report · Decision Letter 2]

10 Feb 2021

Dear Dr. Otto,

We are pleased to inform you that your manuscript 'Bacterial virulence plays a crucial role in MRSA sepsis' has been provisionally accepted for publication in PLOS Pathogens.

Best regards,

Michael R. Wessels

Section Editor

PLOS Pathogens

Michael Wessels

Section Editor

PLOS Pathogens

Kasturi Haldar

Editor-in-Chief

PLOS Pathogens

orcid.org/0000-0001-5065-158X

Michael Malim

Editor-in-Chief

PLOS Pathogens

orcid.org/0000-0002-7699-2064
---

## [Editor Report · Acceptance letter]

18 Feb 2021

Dear Dr. Otto,

We are delighted to inform you that your manuscript, "Bacterial virulence plays a crucial role in MRSA sepsis," has been formally accepted for publication in PLOS Pathogens.

Best regards,

Kasturi Haldar

Editor-in-Chief

PLOS Pathogens

orcid.org/0000-0001-5065-158X

Michael Malim

Editor-in-Chief

PLOS Pathogens

orcid.org/0000-0002-7699-2064